# Evaluating Students’ COVID-19 Knowledge, Attitudes and Practices (COVKAP) during the COVID-19 Pandemic

**DOI:** 10.3390/pharmacy10020046

**Published:** 2022-04-18

**Authors:** David R. Axon, Alina Cernasev, Meghana Desai, Sharon E. Connor, Lauren J. Jonkman, M. Chandra Sekar

**Affiliations:** 1Department of Pharmacy Practice & Science, College of Pharmacy, University of Arizona, 1295 N. Martin Ave., Tucson, AZ 85721, USA; axon@pharmacy.arizona.edu; 2Department of Clinical Pharmacy and Translational Science, College of Pharmacy, University of Tennessee Health Science Center, 301 S. Perimeter Park Dr., Suite 220, Nashville, TN 37211, USA; 3Health Analytics Network, Pittsburgh, PA 15237, USA; desamegh@gmail.com; 4Department of Pharmacy and Therapeutics, University of Pittsburgh School of Pharmacy, 33501 Terrace St, Pittsburgh, PA 15261, USA; sconnor@pitt.edu (S.E.C.); ljf1@pitt.edu (L.J.J.); 5College of Pharmacy, University of Findlay, Findlay, OH 45840, USA; sekar@findlay.edu

**Keywords:** pharmacy, COVID-19, academia, student pharmacist, education

## Abstract

The COVID-19 pandemic led to significant disruption in students’ lives through lockdowns, restricted movement, remote instruction, and mixed information. Therefore, this study aimed to capture the knowledge, attitudes, and practices of student pharmacists during 2020–2021. A 43-item COVID-19 Knowledge, Attitudes, and Practices (COVKAP) survey previously developed was administered at four schools of pharmacy across the U.S. during Fall 2020 and Spring 2021. A total of 418 responses were analyzed from graduating classes of 2021–2024. There were no significant differences in correct COVID-19 knowledge responses across the four graduating years. Respondents’ attitudes around COVID-19 were homogenous with the exception for their belief in their preparedness to administer COVID-19 vaccines. Respondents reported wearing masks daily (76.8%), infrequently visiting restaurants (82.1%), practicing social distancing daily (45.7%), and referring to medical journals for information (72%). In conclusion, during the pandemic, student pharmacists experienced significant changes in their academic lives. Their knowledge and subsequent attitudes and practices were consistent with the state of evidence during Fall 2020 and Spring 2021. Subsequently, as newer information has emerged, the authors suggest that the COVKAP survey may be modified and administered frequently to address student needs and concerns as the pandemic evolves.

## 1. Introduction

The COVID-19 pandemic has led to severe disruption in education globally and in the United States (U.S.). At the beginning of the pandemic in 2020, when local and national governments announced lockdowns, many educational institutions had to rapidly transition to virtual/remote learning despite little to no training and infrastructure to support student needs. As a result, millions of students had to rapidly adapt, often with little support, to a myriad of uncertainties not least of which was their educational environment. Professional healthcare programs that include components of experiential learning had to adjust rapidly to provide didactic courses and rotations that support learning and educational progress in a virtual format, stretching institutional capacity [1]. Universities and Schools/Colleges of Pharmacy continued to provide virtual education in 2021 and, sometimes beyond.

In the U.S., pharmacy education is regulated through the Accreditation Council on Pharmacy Education (ACPE) that requires 1440 hours of advanced pharmacy practice experiences (APPE). The majority of all APPE hours should be in direct patient care; however, non-patient care electives can also be used to prepare students to become practice-ready and to explore areas of interest [2]. Conducting remote instruction of didactic learning is challenging, however, questions around how pharmacy students would cope with remote experiential learning and whether such change would be equivalent to in-person learning remain to be answered. Nevertheless, it is important to address some of these questions—both from the standpoint of instruction and internalized learning along with addressing the suitability of such modalities in the event of future disruptions.

Recent studies in the literature have evaluated the impact of remote learning on students’ mental health and well-being. For example, a multi-center cross-sectional study that assessed medical students across the U.S., perceptions of COVID-19 on the impact of their education, psychological well-being, obligations, and risk of contracting the virus [3] found that most medical students were willing to accept risks and continue in-person rotations (61.3%), and they reported that their learning was hindered by remote rotations (74.7%) [3].

Wallace et al., (2021) evaluated nursing students’ perceptions of transition to virtual didactic courses during the Spring 2020 semester [4] and reported that the abrupt change in delivery to online courses provided an unpredictable amount of new barriers for students to overcome such as technological issues, and engagement in virtual activities [4]. The authors highlighted that superior communication skills were needed to alleviate stressors within the bounds of learning [4]. Thornton et al., (2021) analyzed nursing students’ perceptions at the beginning of the pandemic and reported distress and nervousness regarding the high transmission rate of the SARS-CoV-2 virus, and concern for others, especially vulnerable individuals, including health care workers [5].

In yet another study conducted at Marshall University, Attarabeen et al., (2021) evaluated perceived stress among student pharmacists in adapting to remote learning due to the COVID-19 pandemic [6]. The findings showed no statistically significant data for perceived stress and the authors acknowledged that these results may not represent other pharmacy schools [6].

Impact on mental health: Emerging reports in the literature point to the dramatic effect of the pandemic on individuals’ mental health [7]. For example, the lockdown in April 2020 has been attributed to a 1000% surge in the U.S. mental health hotline usage [8]. Reports from medical clinics have shown an increased number of suicides and deaths attributed to exceedingly poor mental health [9] and a surge in substance use disorders [10]. A two-year longitudinal study of college students in the U.S. reported that students were physically inactive and presented higher levels of anxiety and depression in Fall 2020 [11]. Browning et al., (2021) conducted a cross-sectional survey at six U.S. undergraduate schools and illustrated differences in emotional distress and worrying by gender and age [12]. The study reported reduced motivation to learn, as well as increased anxiety and stress among undergraduate students [12]. In southeastern U.S., Charles et al., (2021) assessed mental health and alcohol use by students in Spring 2020, and reported greater negative outcomes (anger, alcohol use, and mania) by the study’s predominantly female respondents [13]. In a cross sectional survey, Lee et al., (2021) surveyed medical students across the U.S., and reported poor mental health outcomes among white female students [14].

Other stressors: In addition to the challenges of transitioning rapidly to a remote learning environment, additional stressors such as availability of personal protective equipment (PPE) [15], concerns about personal safety and safety of loved ones, and restricted activities and movement may have contributed to the overall anxiety and stress that students experienced during the pandemic.

There is a gap in understanding the magnitude and intensity of the impact of a rapid transition to remote learning amid a pandemic for pharmacy students. As reported by other studies, the impacts on mental health and anxiety need to be examined in greater detail to envision and develop supportive systems and processes that will facilitate and empower students to better adapt and continue their educational journey. Toward this end, our study aimed to collect critical perspectives from pharmacy students regarding their experiences with this rapid transition.

## 2. Materials and Methods

This cross-sectional study utilized a 43-item questionnaire designed specifically for this study and was administered at four schools of pharmacy in the U.S. The development of the COVID-19 Knowledge, Attitudes, and Perceptions (COVKAP) questionnaire was previously published in 2021 [16].

The first section of the questionnaire contained seven descriptive variables, including: graduation year, whether the student was currently working as a pharmacy intern, whether the student worked in any pharmacy setting prior to pharmacy school, how many people (including themselves) lived in their household, how many adults over 65 years of age lived in their household, whether they were considered to be at higher risk of COVID-19 (defined by the CDC as follows: people of any age with certain underlying medical conditions (CKD, COPD, immunocompromised state, obesity, serious heart conditions, sickle cell disease; Type 2 DM), and whether any family members were considered to be at higher risk of COVID-19 (as defined above).

The second section of the questionnaire asked students to respond to 11 knowledge items using a five-point Likert Scale (Strongly disagree to Strongly agree). Students were also asked to indicate the clinical symptoms of COVID-19 from the following list: fever, cough, fatigue, myalgia, loss of taste. The third section of the questionnaire asked students to respond to 15 attitudinal items. The final section of the questionnaire asked students to respond to nine items about their practices during COVID-19.

An initial draft of the questionnaire was developed by the research team, with further revisions made until the instrument was deemed to have appropriate face validity by the research team.

Inclusion criteria: Student pharmacists in graduating classes of 2021, 2022, 2023, or 2024 enrolled at University of Findlay College of Pharmacy, University of Pittsburgh School of Pharmacy, University of Tennessee of Health science Center (UTHSC), College of Pharmacy, and the University of Arizona R. Ken Coit College of Pharmacy were invited to participate in this study.

The COVKAP survey was administered via Qualtrics^®^ in Fall 2020 at University of Findlay, University of Pittsburgh, and University of Tennessee Health Sciences Center and via REDCap (Research Electronic Data Capture) in Spring 2021 at University of Arizona. This study was approved by each institution involved in the study: University of Tennessee Health Science Center (IRB 20-0756-XM; 11 August 2020), University of Pittsburgh (IRB 20090154; 6 September 2020), University of Findlay (IRB 1482, 9 October 2020), and University of Arizona Human Subjects Protection Program (protocol #2021-006-PHPR, 2 April 2021).

An email containing information about the study and a link to participate in the online questionnaire was sent to all eligible participants. A reminder email was sent after one week, and a second reminder email was sent after two weeks. Data collection stopped at the end of the third week. Data were imported into Microsoft Excel (version 16.57, Redmond, WA, USA) and SAS (version 9.4, Cary, NC, USA) for analysis.

Demographic items were summarized and compared between graduating years using Chi-squared tests or Fisher’s exact test (as appropriate). Knowledge items were scored as correct or incorrect, and a sum of the number of correct items per person was calculated. Differences were compared between graduating years using Chi-squared tests or Fisher’s exact test (as appropriate). Since the survey was administered to three schools in fall and one school in spring, a supplementary analysis was conducted to compare knowledge between student pharmacists who completed the survey in fall versus those who completed the survey in spring. Attitude items were grouped as strongly agree/agree versus neutral/disagree/strongly disagree and compared between graduating years using Chi-squared tests or Fisher’s exact test (as appropriate). Agreement items were each scored one point and summed to create an agreement score. A median split (≥9, <9) was used to create two levels of this variable, which served as the dependent variable in a multivariable logistic regression model. Each of the knowledge items served as independent variables in the multivariable logistic regression model. Model parsimony was achieved through deletion of non-significant independent variables. Correlations for various items of interest were calculated, and internal consistency was assessed using Cronbach’s alpha.

## 3. Results

A total of 541 surveys were received (Findlay *n* = 106, Pittsburgh *n* = 98, Tennessee *n* = 217, Arizona *n* = 120). This represented an overall response rate of 36% (Overall 541/1493, Findlay 106/148 (71%), Pittsburgh 98/113 (87%), Tennessee 217/720 (30.1%), Arizona 120/512 (23.4%)). A number of 123 surveys were removed due to missing data (Findlay *n* = 32, Pittsburgh *n* = 37, Tennessee *n* = 39, Arizona *n* = 15), which resulted in a final analytical sample of 418 student pharmacists (Findlay *n* = 74, Pittsburgh *n* = 61, Tennessee *n* = 178, Arizona *n* = 105).

All years of graduation were well-represented, with the class of 2021 representing 19.4% and the class of 2024 representing 32.5% of the survey respondents. The majority (≥50%) of respondents were currently working as a pharmacy intern (69.1%) or had worked in a pharmacy setting prior to pharmacy school (66.0%), and were not considered to be at high risk of contracting COVID-19 (88.3%) at the time of the survey. However, over half of respondents indicated they had a family member who was at high risk of contracting COVID-19 (52.9%). The mean number of people living in a household was 2.9 ± 1.5, and the mean number of adults ≥65 years of age per household was 0.1 ± 0.4 (see Table 1).

Overall, the majority (≥50%) of respondents correctly answered nine-out-of-ten knowledge items, with the overall percentage of correct responses ranging from 58.6 to 97.1%. The exception was the item: “Remdesivir is effective in treating COVID-19 patients”, where only 47.4% of respondents correctly answered to the item. A minority of respondents from the graduating class of 2024 correctly answered the item: “Hydroxychloroquine is effective in treating COVID-19 patients” (43.4%). There were statistically significant differences between graduating years for four items, where a greater percentage of respondents in the more advanced years typically had higher knowledge scores than those in lower years. The overall mean knowledge score was 8.0 ± 1.6 (see Table 2).

The supplementary analysis to determine any differences in knowledge between student pharmacists who completed the survey in fall versus those who completed the survey in spring showed differences in knowledge for two items. A greater proportion of students who responded in the spring correctly answered that COVID-19 is the disease caused by SARS-CoV-2 (94.3 versus 86.9% who answered correctly in the fall, *p* = 0.0382). Likewise, a greater proportion of students who responded in the spring correctly answered that Hydroxychloroquine is effective in treating COVID-19 patients 2 (78.1% versus 60.7% who answered correctly in the fall, *p* = 0.0012; see Table 3).

The majority (70.3%) of respondents indicated that all of the five symptoms presented were associated with COVID-19 (fever, cough, fatigue, myalgia, loss of taste), although 17.7% indicated all symptoms except myalgia were associated with COVID-19. Few respondents indicated any other combination of responses (see Table 4).

The majority of respondents (≥50%) agreed or strongly agreed with nine attitude statements and disagreed, strongly disagreed, or were neutral to six items. There were statistically significant differences between graduating class years for five items. Overall, the majority (≥50%) of respondents felt prepared to administer the COVID vaccine, although significantly fewer first-year students (30.9%) felt prepared to administer the COVID vaccine when compared to other graduating class years (*p* < 0.0001). The mean attitude score was 8.7 ± 2.4 (see Table 5 and Figure 1).

The majority of respondents wore a mask in public every day in the past seven days (76.8%) and past month (69.4%), and practiced social distancing every day or almost every day in the past seven days (82.3%) and past month (84.5%). Respondents reported visiting restaurants or public services two or three times in the past seven days (36.8%) and past month (55.3%), respectively (see Figure 2).

The majority of respondents referred to medical-related journals when they had questions about COVID-19 (72.2%), usually avoided social media sources when they had questions about COVID-19 (78.9%), and were not routinely conducting COVID-19 testing at the drive-through pharmacy (90.0%) (see Figure 3).

The multivariable logistic model demonstrated that two knowledge items (“Remdesivir is effective in treating COVID-19 patients” and “Use of personal protective equipment (e.g., masks) can protect individuals from getting infected”) were significantly associated with higher attitude scores (i.e., an attitude score greater than nine), after adjusting for all other variables in the model. Respondents who correctly responded to the statement “Remdesivir is effective in treating COVID-19 patients” were associated with significantly higher (greater than nine) attitude scores than those who incorrectly responded to the item. Likewise, respondents who correctly responded to the statement “Use of personal protective equipment (e.g., masks) can protect individuals from getting infected” were also associated with significantly higher (greater than nine) attitude scores than those who incorrectly responded to the item. The multivariable logistic model had a c-statistic of 0.654 and a Wald statistic of 0.0047 (see Table 6).

There were significant correlations between the knowledge item “Hydroxychloroquine is effective in treating COVID-19 patients” and the following three attitude items “I believe that it is important for me to counsel my family and friends on COVID-19” (−0.21340, *p* < 0.0001), “I believe that it is important for pharmacists to counsel the general public on COVID-19” (−0.16702, *p* = 0.0006), and “I believe that patients coming to ask about COVID-19 related questions should ask the physician and not the pharmacist” (0.10362, *p* = 0.0342). There were also significant correlations between the knowledge item “Use of personal protective equipment (e.g., masks) can protect individuals from getting infected” and the following two attitude items “I believe that when a patient picks up a prescription and does not wear a mask, it has a negative impact on my interaction with him/her” (0.40491, *p* < 0.0001) and “I believe that when a patient picks up a prescription and wears a mask, it has a positive impact on my interaction with him/her” (0.41937, *p* < 0.0001).

The internal consistency of items in the survey was assessed using Cronbach’s alpha and found to be 0.60, thereby indicating acceptable internal reliability of the COVKAP survey.

## 4. Discussion

The COVID-19 pandemic was a sudden disruption causing panic, stress, anxiety, and rapid adaptations to online learning and work for many individuals. The impact of the pandemic on different populations in general, and on students, in particular, is yet to be fully revealed. Studies in the emerging literature on this topic have reported increased stress, anxiety, poor learning outcomes, and other undesirable impacts from a rapid transition to virtual or remote learning [1,2,3,4,5,6].

Our study was one of the first to address the knowledge, attitudes, and perceptions of student pharmacists during the COVID pandemic. It was expected that student pharmacists would be generally knowledgeable about the pandemic, would seek information from scientific literature, and thus their attitudes would be shaped by a scientific temper. Student pharmacists reported feeling comfortable on counseling family members and friends on COVID-19 treatments.

Student pharmacists’ attitudes towards PPEs were reflective of the state of knowledge at the time the study was conducted. A surprising finding was that only 69% (56) student pharmacists graduating in 2021 believed that they were adequately prepared to administer the COVID-19 vaccine. This is indeed surprising since it is expected that student pharmacists are trained to vaccinate and have completed their immunization administration requirements by the time of graduation. Subsequent classes of 2022 and 2023 reported better preparation (77 and 66%) respectively. Colleges and Schools of Pharmacy may want to investigate this further with their graduating classes as they prepare to take their board exams and begin their careers as pharmacists.

Due to lockdowns and restrictions, there was a rapid transition to remote learning. Yet, many schools and colleges were not adequately prepared to make this transition. This may have caused additional stress and anxiety as students were trying to adapt to these changes [13,14,15]. In our study of 418 student pharmacists, 49 (11%) reported that they were at high-risk for COVID-19 as defined by the Centers for Disease Control and Prevention. More than half—221 (52%)—reported that they had a family member who was at a higher risk for COVID-19 infections. These additional challenges may have contributed to the general anxiety and stress for students. Academic institutions and administrators may consider developing preparedness and risk mitigation plans for future disruptions based on the lessons learned during the pandemic. Particularly, in the case of pandemics or epidemics, it would be important to identify quickly and effectively those students who would be at higher risk themselves or have family members at higher risk for infections or mortality. This would enable administrators to prioritize steps to mitigate risks to high-risk students and those who may have a high-risk family member. Such risk stratification and mitigation strategies could help university administrators to be prepared to effectively identify and isolate high-risk students, and perhaps reduce transmission(s).

Early information and communication of scientific knowledge is crucial during a pandemic. Student pharmacists demonstrated high overall knowledge of COVID-19 (mean score was 8.0 out of 10, SD = 1.6) indicating that they were aware of the information and (mis)information about COVID-19 at the time of this study. Responses to the attitude questions revealed underlying concerns about availability of PPE and safety measures in pharmacies.

In terms of patient interactions, we asked students two questions about their interactions with patients who did or did not wear masks. Interestingly, 232 (55%) students reported that if patients did not wear a mask, it may have a negative impact on their interactions. Additionally, 244 (58%) students believed that if patients wore masks, it would have a positive impact on their interactions. The opposing questions were asked to test whether students were (a) attentive to the survey when responding, and (b) whether there were differences in their own attitudes towards patients when they wore masks or not. Student pharmacists are increasingly aware of safety and risk parameters and these findings are not unusual. However, there is a need for universities and pharmacy workplaces to consider the impact of their safety measures on student pharmacists, particularly when it comes to interacting with patients and providing direct care.

292 (68%) student pharmacists in the survey reported that they would get vaccinated when available. This is an important finding since vaccine hesitancy has emerged as an important challenge during the pandemic [17]. It may be noted that during Fall 2020, the vaccinations were being formulated and there were questions about storage, refrigeration, efficacy, among others that should be taken into account when considering these results. While this study did not explore facets of vaccine hesitancy, this may be explored in future studies, particularly, since vaccine mandates have been implemented in many universities and workplaces in the past year. University administrators may want to take into account student attitudes regarding vaccinations when developing preparedness policies.

In this study, we aimed to quantify behaviors on mask wearing in public, visiting crowded places, etc. Academic institutions and administrators may want to consider developing mechanisms to rapidly elicit such information during a pandemic or epidemic in order to mitigate risks of transmission. Lastly, academic administrators may want to consider development and deployment of rapid response tools such as this survey to better understand their students’ concerns and needs during rapidly changing emergencies.

### Study Limitations

While the study provides early insights into student pharmacists’ knowledge, attitudes and perceptions regarding COVID while in the midst of the pandemic, the study has certain limitations. Firstly, the study included four schools of pharmacy across four different geographic regions (Midwest, Northeast, and West Coast), however, results of this study are limited in their generalizability and had a response rate of 36%. Secondly, the study was administered during a rapidly evolving pandemic that may have led to change in knowledge and attitudes, however, as each student was only surveyed once, any changes were not captured in this cross-sectional study. Our supplementary analysis indicated a significant difference in knowledge for two items (“COVID-19 is the disease caused by SARS-CoV-2” and “Hydroxychloroquine is effective in treating COVID-19 patients”) between students surveyed in the fall and those surveyed in the spring semester, which may have been due to evolving knowledge. Furthermore, the study used a previously developed instrument that was assessed for face validity by the researchers. Further work to assess other aspects of the instrument’s validity could be conducted in future to make a more refined instrument with greater evidence of validity.

## 5. Conclusions

The COVID-19 pandemic caused enormous disruption in academia and led to rapid adaption of remote learning. Student pharmacists in this study were generally knowledgeable about COVID-19, with a mean score of 8.0 ± 1.6 out of 10. Student pharmacists’ attitudes about COVID-19 varied depending on the statement, and there were some differences in knowledge and attitudes between graduating years. Academic administrators may want to consider risk mitigation strategies and evaluate student pharmacists’ perceived risks in order to make strategic decisions in prioritizing appropriate mitigation measures. Future directions involve evaluating student pharmacists’ preparedness—both for practice and during emergencies—along with exploration of different aspects of pandemics such as vaccine hesitancy, knowledge seeking, and risk-taking behaviors.

## Figures and Tables

**Figure 1 pharmacy-10-00046-f001:**
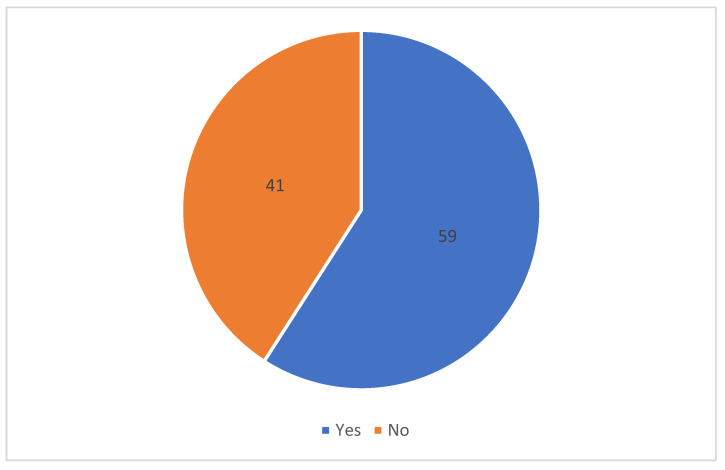
Student pharmacists’ responses to the item: I believe that I am adequately prepared to administer the COVID-19 vaccine to patients (*n* = 418).

**Figure 2 pharmacy-10-00046-f002:**
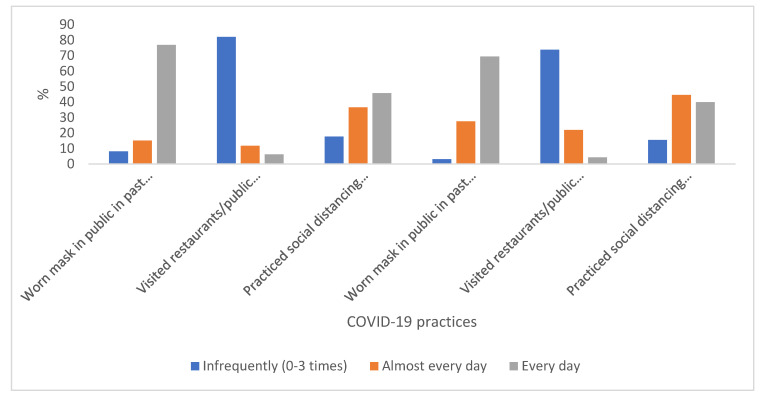
COVID-19 practices among student pharmacists included in the study (*n* = 418).

**Figure 3 pharmacy-10-00046-f003:**
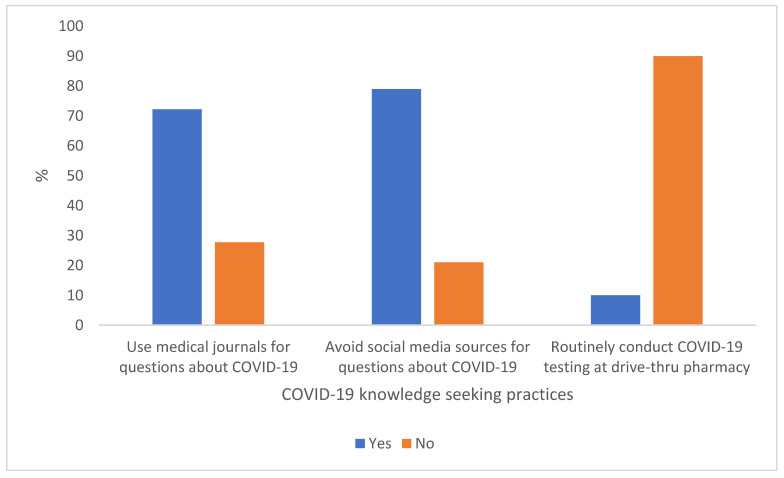
COVID-19 knowledge seeking practices among student pharmacists included in the study (*n* = 418).

**Table 1 pharmacy-10-00046-t001:** Demographic characteristics of student pharmacists included in the study (*n* = 418).

Demographic Characteristics	Graduation Year 2021 (*n* = 81) Frequency (%)	Graduation Year 2022 (*n* = 107) Frequency (%)	Graduation Year 2023 (*n* = 94) Frequency (%)	Graduation Year 2024 (*n* = 136) Frequency (%)	*p*-Value
School					<0.0001
Findlay	16 (21.6)	28 (37.8)	7 (9.5)	23 (31.1)	
Pittsburgh	17 (27.9)	20 (32.8)	6 (9.8)	18 (29.5)	
Tennessee	26 (14.6)	37 (20.8)	45 (25.3)	70 (39.3)	
Arizona	22 (20.1)	22 (20.1)	36 (34.3)	25 (23.8)	
Are you currently working as a pharmacy intern?					<0.0001
Yes	65 (22.5)	94 (32.5)	63 (21.8)	67 (23.2)	
No	16 (12.4)	13 (10.1)	31 (24)	69 (53.5)	
Did you work in any pharmacy setting prior to pharmacy school?					0.9993
Yes	53 (19.2)	71 (25.7)	62 (22.5)	90 (32.6)	
No	28 (19.7)	36 (25.4)	32 (22.5)	46 (32.4)	
Are you considered to be at higher risk in COVID-19 (as defined by the CDC as follows: People of any age with certain underlying medical conditions (CKD, COPD, Immunocompromised state, Obesity, Serious heart conditions, Sickle cell disease; Type 2 DM)?					0.1629
Yes	8 (16.3)	7 (14.3)	14 (28.6)	20 (40.8)	
No	73 (19.8)	100 (27.1)	80 (21.7)	116 (31.4)	
Do you have any family members considered to be at higher risk in COVID-19 (as defined by the CDC as follows: People of any age with certain underlying medical conditions (CKD, COPD, Immunocompromised state, Obesity, Serious heart conditions, Sickle cell disease; Type 2 DM)?					0.9202
Yes	44 (19.9)	55 (24.9)	52 (23.5)	70 (31.7)	
No	37 (18.8)	52 (26.4)	42 (21.3)	66 (33.5)	
How many people including yourself are living in your household? Mean (SD)	3.0 (1.5)	3.0 (1.6)	2.8 (1.3)	2.9 (1.5)	0.7990
How many adults over 65 years living in your household? Mean (SD)	0.1 (0.4)	0.1 (0.5)	0.1 (0.3)	0.1 (0.4)	0.7363

CDC = Centers for Disease Control and Prevention; CKD = chronic kidney disease, COPD = chronic obstructive pulmonary disease; DM = diabetes mellitus; SD = standard deviation.

**Table 2 pharmacy-10-00046-t002:** Knowledge of COVID-19 among student pharmacists included in the study (*n* = 418).

Knowledge Statement	Graduation Year 2021 (*n* = 81) Correct N (%)	Graduation Year 2022 (*n* = 107) Correct N (%)	Graduation Year 2023 (*n* = 94) Correct N (%)	Graduation Year 2024 (*n* = 136) Correct N (%)	*p*-Value
COVID-19 is the disease caused by SARS-CoV-2	80 (98.8)	94 (87.9)	82 (87.2)	115 (84.6)	0.0126
Hydroxychloroquine is effective in treating COVID-19 patients	68 (84.0)	74 (69.2)	71 (75.5)	59 (43.4)	<0.0001
Remdesivir is effective in treating COVID-19 patients	47 (58.0)	59 (55.1)	35 (37.2)	57 (41.9)	0.0082
COVID-19 develops only among the elderly	78 (96.3)	100 (93.5)	90 (95.7)	127 (93.4)	0.7213
Eating or being in contact with wild animals would result in SARS-CoV-2 infections in humans	51 (63.0)	60 (56.1)	55 (58.5)	79 (58.1)	0.8177
COVID-19 spreads through respiratory droplets of infected individuals	79 (97.5)	104 (97.2)	91 (96.8)	132 (97.1)	1.0000
COVID-19 can spread asymptomatically	78 (96.3)	101 (94.4)	91 (96.8)	130 (95.6)	0.8745
Use of personal protective equipment (e.g., masks) can protect individuals from getting infected	77 (95.1)	89 (83.2)	84 (89.4)	109 (80.2)	0.0123
To prevent COVID-19 infection, individuals should avoid crowded places	78 (96.3)	96 (89.7)	91 (96.8)	123 (90.4)	0.0932
If an individual travels to another state or region where COVID-19 cases are high, they should self-isolate upon their return	63 (77.8)	83 (77.6)	77 (81.9)	107 (78.7)	0.8749

**Table 3 pharmacy-10-00046-t003:** Knowledge of COVID-19 among student pharmacists included in the study stratified by semester the survey was administered (*n* = 418).

Knowledge Statement	Fall Semester (*n* = 313) Correct N (%)	Spring Semester (*n* = 105) Correct N (%)	*p*-Value
COVID-19 is the disease caused by SARS-CoV-2	272 (86.9)	99 (94.3)	0.0382
Hydroxychloroquine is effective in treating COVID-19 patients	190 (60.7)	82 (78.1)	0.0012
Remdesivir is effective in treating COVID-19 patients	153 (48.9)	45 (42.9)	0.2847
COVID-19 develops only among the elderly	296 (94.6)	99 (94.3)	0.9124
Eating or being in contact with wild animals would result in SARS-CoV-2 infections in humans	181 (57.8)	64 (61.0)	0.5737
COVID-19 spreads through respiratory droplets of infected individuals	303 (96.8)	103 (98.1)	0.7381
COVID-19 can spread asymptomatically	299 (95.5)	101 (96.2)	1.0000
Use of personal protective equipment (e.g., masks) can protect individuals from getting infected	268 (85.6)	91 (86.7)	0.7904
To prevent COVID-19 infection, individuals should avoid crowded places	291 (93.0)	97 (92.4)	0.8393
If an individual travels to another state or region where COVID-19 cases are high, they should self-isolate upon their return	241 (77.0)	89 (84.8)	0.0912

**Table 4 pharmacy-10-00046-t004:** Knowledge of COVID-19 clinical symptoms among student pharmacists included in the study (*n* = 418).

COVID-19 Clinical Symptoms Selected:	*n* (%)
All (fever, cough, fatigue, myalgia, loss of taste)	294 (70.3)
All except myalgia	74 (17.7)
All except fatigue	1 (0.2)
All except loss of taste	5 (1.2)
All except cough	10 (2.4)
All except fever	1 (0.2)
All except fatigue and myalgia	12 (2.9)
All except cough and myalgia	9 (2.2)
All except cough and loss of taste	2 (0.5)
All except myalgia and loss of taste	5 (1.2)
All except fever and myalgia	1 (0.2)
Just cough and fatigue	1 (0.2)
Just fever and fatigue	1 (0.2)
Just fever and cough	2 (0.5)

**Table 5 pharmacy-10-00046-t005:** Attitudes about COVID-19 among student pharmacists included in the study (*n* = 418).

Attitude Statement	Graduation Year 2021 (*n* = 81) Strongly Agree/Agree N (%)	Graduation Year 2022 (*n* = 107) Strongly Agree/Agree N (%)	Graduation Year 2023 (*n* = 94) Strongly Agree/Agree N (%)	Graduation Year 2024 (*n* = 136) Strongly Agree/Agree N (%)	*p*-Value
I believe that COVID-19 is under control in the United States	7 (8.6)	12 (11.2)	4 (2.3)	26 (19.1)	0.0046
I believe that adequate PPEs are available for all healthcare workers	29 (35.8)	34 (31.8)	23 (24.5)	52 (38.2)	0.1618
I believe that adequate PPEs are available for members of the general public	30 (37.0)	35 (32.7)	36 (38.3)	56 (41.2)	0.6014
I believe that information about COVID-19 testing and prevention should be available at all pharmacies	75 (92.6)	91 (85.1)	87 (92.6)	129 (94.9)	0.0492
I believe that pharmacists have adequate PPEs to protect themselves	39 (48.2)	44 (41.1)	40 (42.6)	73 (53.7)	0.1946
I believe that pharmacies have adequate protections to prevent infections	38 (46.9)	42 (39.3)	35 (37.2)	76 (55.9)	0.0166
I believe that pharmacists should participate in COVID-19 immunizations	70 (86.4)	87 (81.3)	85 (90.4)	126 (92.7)	0.0449
I believe that pharmacists should participate in public health taskforces	77 (95.1)	98 (91.6)	84 (89.4)	126 (92.7)	0.5640
I believe that it is important for me to counsel my family and friends on COVID-19	72 (88.9)	89 (83.2)	85 (90.4)	114 (83.8)	0.3403
I believe that it is important for pharmacists to counsel the general public on COVID-19	75 (92.6)	97 (90.7)	90 (95.7)	130 (95.6)	0.3377
I believe that patients coming to ask about COVID-19 related questions should ask the physician and not the pharmacist	2 (2.5)	5 (4.7)	5 (5.3)	12 (8.8)	0.2360
I believe that when a patient picks up a prescription and does not wear a mask, it has a negative impact on my interaction with him/her	45 (55.6)	57 (53.3)	59 (62.8)	71 (52.2)	0.4198
I believe that when a patient picks up a prescription and wears a mask, it has a positive impact on my interaction with him/her	45 (55.6)	58 (54.2)	64 (68.1)	77 (56.6)	0.1831
I believe that I should get vaccinated when the COVID-19 vaccine becomes available	54 (66.7)	71 (66.4)	73 (77.7)	92 (67.7)	0.2651
I believe that I am adequately prepared to administer the COVID-19 vaccine to patients	56 (69.1)	83 (77.6)	66 (70.2)	42 (30.9)	<0.0001

PPEs = personal protective equipment.

**Table 6 pharmacy-10-00046-t006:** Association between COVID-19 knowledge items and higher (≥9) attitude scores as indicated by multivariable logistic regression among student pharmacists included in the study (*n* = 418).

Knowledge Items (Independent Variables)	Odds Ratio (95% CI)
COVID-19 is the disease caused by SARS-CoV-2	1.561 (0.815, 2.991)
Hydroxychloroquine is effective in treating COVID-19 patients	0.926 (0.598, 1.435)
Remdesivir is effective in treating COVID-19 patients	**1.657 (1.102, 2.491)**
COVID-19 develops only among the elderly	1.042 (0.390, 2.785)
Eating or being in contact with wild animals would result in SARS-CoV-2 infections in humans	0.956 (0.628, 1.455)
COVID-19 spreads through respiratory droplets of infected individuals	4.200 (0.733, 24.082)
COVID-19 can spread asymptomatically	0.602 (0.173, 2.095)
Use of personal protective equipment (e.g., masks) can protect individuals from getting infected	**2.540 (1.298, 4.969)**
To prevent COVID-19 infection, individuals should avoid crowded places	1.276 (0.486, 3.350)
If an individual travels to another state or region where COVID-19 cases are high, they should self-isolate upon their return	1.246 (0.738, 2.102)

CI = confidence interval. C-statistic = 0.654. Wald statistic = 0.0047. **Bold** odds ratios indicate significant association between knowledge and attitude scores.

## Data Availability

The data presented in this study are available on request from the corresponding author. The data are not publicly available due to privacy concerns.

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
