# Peer review of "Evaluating Students’ COVID-19 Knowledge, Attitudes and Practices (COVKAP) during the COVID-19 Pandemic"

_pharmacy, 2022, doi:10.3390/pharmacy10020046_

Round 1

Reviewer 1 Report

I enjoyed reading the manuscript. It explores students' COVID-19 knowledge, attitudes and practices using 43-item survey designed specifically for this study (the development of the questionnaire was previously described and published in Pharmacy 2021; 9(4)) . The paper is very well written, study having appropriate design and very well written methodology.   I have the only minor comments, intended to improve the quality.

1) Could you please provide the response rate

2) The results of multivariable logistic model are demanding for interpretation and authors missed to explain them. Please, add the discussion on those results/ interpret them .

3) line 125 - please delete "to respond" (written twice)

Author Response

I enjoyed reading the manuscript. It explores students' COVID-19 knowledge, attitudes and practices using 43-item survey designed specifically for this study (the development of the questionnaire was previously described and published in Pharmacy 2021; 9(4)) . The paper is very well written, study having appropriate design and very well written methodology.   I have the only minor comments, intended to improve the quality.

  • Could you please provide the response rate

Thank you for your comments on our paper. We have added the response rate to the results section

  • The results of multivariable logistic model are demanding for interpretation and authors missed to explain them. Please, add the discussion on those results/ interpret them.

We have simplified our interpretation of the logistic regression results text by removing the odds ratios from the text and added bold to indicate significant associations in the corresponding table.

  • line 125 - please delete "to respond" (written twice)

Thank you for catching this. We have removed the duplicate “to respond”.

Reviewer 2 Report

Thank you for the opportunity to review, and your work on this subject.

Methods/Results: I like the survey tool, but I would like a better explanation of how it was "deemed" appropriate as there are ways to validate tools beyond the research team. I would like to see further delineation in your methods on when the surveys were deployed beyond just Fall and Spring. It is odd to have the surveys sent at different times especially with the ever changing research with COVID (ie vaccines not approved in Fall, but readily available to public in April/May). If all Fall surveys were sent at the same time, then I would like a secondary analysis done on the knowledge items to make sure that Fall respondents are not significantly different than Fall. If fall surveys were sent in all different months, then I have more concerns about the methods. The authors have not listed the survey response rate, but this looks like a small response rate and that should be noted in the limitations. 

Discussion section, remove second paragraph about two semesters of knowledge as you only surveyed a student group one time and are not tracking those items over two semesters. I feel that surveying students at different times is a limitation to this study as it is not the study design, but rather a response to a delayed IRB. The discussion section also talks about vaccine hesitancy and lack of training, but the vaccine was not approved and there were no formal guidance on use until December 2020/Jan 2021 so that is partially why I would like to see a breakdown of Fall vs Spring and know when the Fall surveys were deployed. I also think PPE and treatment questions likely had different scientific evidence during the survey deployment period. 

Based on the results, the authors should make a more clear statement in the conclusion on what these numbers mean. Do statistically significant differences make a difference to administrators? Is there a cut off of % of students that response yes/no to questions that should require action?

Author Response

Thank you for the opportunity to review, and your work on this subject.

Methods/Results: I like the survey tool, but I would like a better explanation of how it was "deemed" appropriate as there are ways to validate tools beyond the research team. I would like to see further delineation in your methods on when the surveys were deployed beyond just Fall and Spring. It is odd to have the surveys sent at different times especially with the ever changing research with COVID (ie vaccines not approved in Fall, but readily available to public in April/May). If all Fall surveys were sent at the same time, then I would like a secondary analysis done on the knowledge items to make sure that Fall respondents are not significantly different than Fall. If fall surveys were sent in all different months, then I have more concerns about the methods. The authors have not listed the survey response rate, but this looks like a small response rate and that should be noted in the limitations. 

Thank you for your valuable feedback. The research team assessed the face validity and content validity of this instrument through an iterative process of reviewing the questions with all the team members. A description of how the survey was developed is provided in our previous publication (https://www.mdpi.com/2226-4787/9/4/159/review_report). We have added to the limitations and future work section that further work could be done to further assess the validity of the tool.

The goal was to capture student knowledge and attitudes during the academic year. Each school started the fall semester at different times, so there were some differences in when the survey was administered during the fall. We have added the results of a supplementary analysis of student knowledge between the fall and spring semesters (new Table 3), along with some complementary text. There were two variables where there were significant differences in knowledge between students who responded in the fall and students who responded in the spring. We have commented on this in the limitations.

We have added the survey response rate and mentioned this in the limitations.

Discussion section, remove second paragraph about two semesters of knowledge as you only surveyed a student group one time and are not tracking those items over two semesters. I feel that surveying students at different times is a limitation to this study as it is not the study design, but rather a response to a delayed IRB. The discussion section also talks about vaccine hesitancy and lack of training, but the vaccine was not approved and there were no formal guidance on use until December 2020/Jan 2021 so that is partially why I would like to see a breakdown of Fall vs Spring and know when the Fall surveys were deployed. I also think PPE and treatment questions likely had different scientific evidence during the survey deployment period. 

We have revised the sentence about two semesters of knowledge. We feel the remainder of the paragraph is still relevant to the manuscript and have retained it. The question of attitudes towards vaccines is an important one and has become one of the largest factors in addressing the pandemic. It was deemed important early on during survey development to identify attitudes around risky behaviors. Pharmacists are public health providers and authorization to immunize was a significant public health achievement, particularly for pharmacists. Therefore, we believe that it is important to identify student pharmacists’ attitudes towards vaccinations that may help predict behaviors.

Based on the results, the authors should make a more clear statement in the conclusion on what these numbers mean. Do statistically significant differences make a difference to administrators? Is there a cut off of % of students that response yes/no to questions that should require action.

We have added extra clarity to our conclusion. We believe that our survey is limited in its conclusions and providing cutoffs would be beyond the aim and results of the survey. We have mentioned that administrators may want to consider a version of this survey or similar tools to rapidly collect information in evolving disasters and emergencies that may guide decision making.

Reviewer 3 Report

Thank you for the opportunity to review your manuscript. Please see my attached report for comment and occasional suggestions. There were also a very few word choices used that may perhaps be altered to enhance the scholarly tone of the work.

Author Response

Thank you for your comments on our paper. We agree with the suggestions made and have made edits to the abstracts, introduction, and discussion sections.

Round 2

Reviewer 2 Report

Thank you for the modifications that were provided. I think these add to the value of this paper and will allow for better applicability. Thank you for adding the supplementary analysis and limitations. 

On page 9 & 10, when discussing the logistic regression model, it does not appear that the data is presented. It looks like Table 6 is presenting it, but I am not following how the 2 statements bolded were selected.

Table 6, Why is this NOT bolded and discussed as a higher correlation? "COVID-19 spreads through respiratory droplets of infected individuals" 4.200 (0.733, 24.082)

Author Response

Thank you for your additional review of the paper. The data for the logistic regression results are presented in Table 6. The two bolded results indicate that there is a significant association between knowledge and attitude score. This is determined by the odds ratio for these statements not crossing 1. We have added how to interpret significance in odds ratios to the methods section. For the statement, “COVID-19 spreads through respiratory droplets of infected individuals”, the confidence interval does not include 1 which indicates there was no significant association between this knowledge item and higher attitude score. Therefore, it is not bolded.